# From Federal Heterogeneity to Reproducible Analytics: A Provenance-Aware Knowledge Graph for Cross-Portal Comparability of German Open Government Data

Position and Vision Paper

Florian Hahn[1,*], Michael Martin[1] and Eric Engels[2]

[1]*Chemnitz University of Technology, Professorship of Data Management, Chemnitz, Germany*
[2]*Hessian Ministry for Digitalisation and Innovation, Wiesbaden, Germany*

### Abstract

Cross-portal analyses of open government data in Germany are hindered by federal heterogeneity. Each of Germany's 16 federal states operates distinct publication setups, organizational structures, and topic taxonomies, and these can shift over time due to administrative reforms and political terms. Consequently, geographical and thematic questions, such as the identification of the ministries that published which types of datasets under which category labels in which period, remain challenging to answer reproducibly. This is particularly evident when various political stakeholders undertake a comparative analysis of their ministry or subject area against analogous ministries in federal states. The position and vision paper proposes a knowledge graph that (i) captures portal- and state-specific category schemes as versioned resources, (ii) links publishers to persistent authority identifiers, and (iii) records extraction and mapping provenance for auditability. Rather than proposing a new foundational ontology or mapping language, the contribution is a reusable construction pattern for provenance-aware and temporally scoped knowledge graph construction in federated open data settings. The approach is aligned with the Knowledge Graph Construction Workshop's focus on mapping-based knowledge graph construction, workflows, and provenance-aware pipelines.

### Keywords

Knowledge Graph Construction, Open Government Data, Dataset categorization, Provenance, Metadata, RML

## 1. Motivation

Germany is a federal republic of 16 states with their own administrative structures and responsibilities [1]. In practice, this leads to a fragmented Open Data landscape: portals differ in software stacks (often CKAN-based), metadata profiles (for Germany commonly DCAT-AP.de), publisher naming conventions, and category vocabularies [2, 3]. Even when datasets are harvested into aggregators, the semantics of state-specific categories and the evolving organizational context (ministries, subordinate agencies, reorganizations) are typically not represented in a way that supports historical analysis.

This directly motivates a knowledge graph construction problem: build a graph that is not only an integrated catalog of datasets and publishers, but also a longitudinal representation of changing category schemes and organizational assignments, with explicit provenance and temporal validity.

## 2. Problem statement

The concrete research gap addressed here is the missing, reproducible alignment between three evolving layers:

- **Portal layer:** portal instances and their dataset metadata, usually accessible via CKAN Action APIs or DCAT feeds [4, 5].

*KGCW'26: 7th International Workshop on Knowledge Graph Construction, May 10th, 2026, Dubrovnik, Croatia*
*Corresponding author.

✉ florian.hahn@informatik.tu-chemnitz.de (F. Hahn)
🌐 https://www.tu-chemnitz.de/informatik/dm/team/fh.php.en (F. Hahn)
🆔 0009-0008-1126-9319 (F. Hahn); 0000-0003-0762-8688 (M. Martin)

- **Organization layer:** publishers, ministries, and agencies, frequently represented as strings and changing due to renamings and reorganizations.
- **Category layer:** portal-specific thematic taxonomies that can be renamed, split, merged, or replaced across time.

Without explicitly modeling time and provenance, comparisons such as category drift, publisher reassignment, or cross-state thematic coverage become non-auditable and hard to replicate. A purely static mapping of categories to a fixed vocabulary also risks masking historically valid structures. The Integrated Authority File (Gemeinsame Normdatei, GND) provides dereferenceable identifiers that can serve as anchors for public-sector entities; one record is shown as a small graph in the GND linked-data view [6]. In addition, relevant contextual information is often published in non-machine-readable formats (e.g., PDFs), which complicates automated extraction. In ongoing work, we explore how such sources (e.g., coalition agreements) can be integrated as additional evidence for organizational change. An overview of all 16 federal states and their coalition agreements is being compiled.

## 3. Vision

We propose a temporal and provenance-aware knowledge graph that treats (i) portal categories and (ii) organizational structures as first-class, time-scoped resources. The key difference to many catalog integration efforts is that the graph explicitly represents how administrative entities and their thematic assignments evolve over time; beyond the use case, we contribute a reusable KGC pattern combining time-scoped organization versions, explicit ORG change events, provenance-tracked mappings with explicit inference marking, and SHACL-based validation. While Germany serves as a demanding federal case, the construction pattern is intended for any decentralized OGD setting in which organization structures, publisher identities, and category schemes evolve over time.

### 3.1. Time-scoped organizational reality

Instead of treating a ministry or agency name as the identity itself, we distinguish between a stable organization as an identity anchor and its time-scoped organizational versions. A stable organization denotes one administratively continuous public-sector entity across multiple valid-time states. This supports longitudinal queries such as which ministry name was valid in a given period, how responsibilities shifted, and which sub-units existed under which superior organization.

### 3.2. Explicit organizational change events

Reorganizations, renamings, splits, and merges are modeled as explicit change events rather than as overwritten fields. In this design, renamings preserve the stable organization, whereas splits and mergers terminate predecessor continuity and create successor organizations with new stable identifiers linked through typed change events. We adopt a change-event representation aligned with the W3C ORG vocabulary to connect predecessor and successor organizations and to attach temporal metadata to the change.

### 3.3. Category assignments with EU Data Theme alignment

Category and theme information is represented as SKOS concepts, and assignments are time-scoped. Where possible, portal-specific categories are aligned to a stable cross-portal vocabulary, in our case the EU Publications Office Data theme authority table [7]. This preserves local schemes while enabling cross-portal aggregation and comparison.

### 3.4. Reproducible, declarative construction with provenance

Every snapshot, transformation, and mapping result is recorded with provenance so that a past state can be reconstructed. We use PROV-O for process provenance [8] and OWL-Time for temporal intervals [9]. Declarative mapping rules (RML) are used to keep transformations explicit and re-executable [10, 11].

## 4. Worked example: organization history from a tabular template

To make the intended modeling concrete, we provide a worked example derived from a small tabular template (`organRdfFromCSV.ods`; see Appendix A). The template contains (a) attributes (header row), (b) datatypes, and (c) example rows describing a ministry-like organization name, its administrative level (`organType` as an enum), locations, validity dates, optional hierarchical relations, and optional predecessor/successor fields.

From each example row we generate the core graph artifacts needed for representing organizational continuity, change, and thematic assignment:

- a time-scoped **organization version** resource describing the state of an organization during a validity interval,
- a stable **organization** resource used as the anchor for identities and relationships,
- an optional **change event** resource connecting predecessor and successor organizations, and
- a time-scoped **category assignment** that can be aligned to EU Data Theme.

A practical issue visible in the example rows is that predecessor/successor strings can be inconsistent with the validity intervals. To remain reproducible and auditable, the graph records both the raw input strings and the chosen change-event direction. If a direction is inferred from dates rather than taken verbatim from the predecessor/successor columns, the event is explicitly annotated as inferred to avoid presenting the inference as a fact.

## 5. Proposed data model

We adopt DCAT as the base for catalogs and datasets [5], and complement it with an organization-history and category-alignment layer plus provenance and time. The resulting KG covers portal snapshots, datasets and their publishers, stable organizations and time-scoped organization versions, typed organizational change events, portal-specific categories, cross-portal alignments, and provenance records for extraction and mapping.

### 5.1. Modules

- **State module:** a lightweight representation of federal states (using official keys and abbreviations) as contextual anchors [1].
- **Organization module:** stable organizational resources (`org:Organization`) with human-readable labels and hierarchical relations, complemented by time-scoped organization versions.
- **Change module:** explicit reorganization events (`org:ChangeEvent`) that connect original and resulting organizations, carry temporal metadata, and are typed at least as renaming, split, merger, or responsibility transfer/reorganization.
- **Category module:** portal-specific categories as `skos:Concept` plus time-scoped assignments, optionally aligned via `skos:exactMatch` to EU Data Theme [7].
- **Provenance and time module:** snapshot activities and generated artifacts captured using PROV-O [8] and validity intervals using OWL-Time [9].

### 5.2. Illustrative RDF example

Listing 1 illustrates the core pattern used in the worked example; the companion repository is referenced in Appendix A. It shows (i) a stable organization, (ii) a time-scoped organization version with a validity interval, (iii) a change event linking predecessor and successor organizations, and (iv) a category assignment aligned to EU Data Theme. The example uses the example.org namespace for illustration.

Listing 1: Illustrative Turtle snippet for time-scoped organization versions, change events, and EU theme alignment.

```
@prefix org:  <http://www.w3.org/ns/org#> .
@prefix prov: <http://www.w3.org/ns/prov#> .
@prefix time: <http://www.w3.org/2006/time#> .
@prefix dct:  <http://purl.org/dc/terms/> .
@prefix skos: <http://www.w3.org/2004/02/skos/core#> .
@prefix ex:   <https://example.org/sodorg/> .
@prefix eu:   <http://publications.europa.eu/resource/authority/data-theme/> .
@prefix sod:  <https://example.org/sodorg/ontology/> .

ex:org/ministerium-energiewende a org:Organization ;
  skos:prefLabel "Ministerium für Energiewende"@de ;
  skos:altLabel "Ministry for Energy"@en .
ex:organVersion/ministerium-energiewende_2017-02-28 a sod:OrganVersion ;
  sod:organ ex:org/ministerium-energiewende ;
  sod:organType sod:OrganType_federalState ;
  dct:spatial ex:place/Kiel ;
  dct:temporal ex:interval/2017-02-28_to_2021-02-28 ;
  prov:wasGeneratedBy ex:activity/template-mapping_2026-02-17 .
ex:changeEvent/2017-02-28 a org:ChangeEvent, prov:Activity ;
  org:originalOrganization ex:org/ministerium-landwirtschaft ;
  org:resultingOrganization ex:org/ministerium-energiewende ;
  prov:endedAtTime "2017-02-28T00:00:00Z"^^<http://www.w3.org/2001/XMLSchema#dateTime> ;
  sod:directionInferred "true"^^<http://www.w3.org/2001/XMLSchema#boolean> ;
  sod:inferenceNote "Direction derived from validity dates; raw predecessor/successor strings
      preserved."@en .
ex:categoryAssignment/ministerium-energiewende_2017-02-28 a sod:CategoryAssignment ;
  sod:assignedTo ex:organVersion/ministerium-energiewende_2017-02-28 ;
  sod:category ex:category/Energie .
ex:category/Energie a skos:Concept ;
  skos:prefLabel "Energie"@de ;
  skos:exactMatch eu:ENER .
ex:interval/2017-02-28_to_2021-02-28 a time:Interval ;
  time:hasBeginning [ a time:Instant ;
    time:inXSDDateTime "2017-02-28T00:00:00Z"^^<http://www.w3.org/2001/XMLSchema#dateTime> ] ;
  time:hasEnd [ a time:Instant ;
    time:inXSDDateTime "2021-02-28T00:00:00Z"^^<http://www.w3.org/2001/XMLSchema#dateTime> ] .
```

## 6. Construction workflow

The workflow is snapshot-driven and designed for reproducibility and auditability:

1. **Template ingestion:** read the tabular template with declared attribute names and datatypes and convert each data row into an intermediate record.
2. **Stable identity minting:** mint stable IRIs for organizations using normalized names together with administrative context and continuity rules; renamings preserve identity, whereas splits and mergers create successor identities linked by change events.

3. **Time-scoped versions:** mint one `sod:OrganVersion` per row using a version IRI independent of the end date; `organDateFrom` and `organDateTo` are represented in the validity interval.

4. **Hierarchy:** if `hierarchicalOrganSuperior` or `hierarchicalOrganSubordinate` are present, create `org:subOrganizationOf` and `org:hasSubOrganization` relations.

5. **Change events:** if predecessor/successor hints exist, create `org:ChangeEvent` resources. If the template direction is inconsistent with validity intervals, the direction is derived from dates and marked as inferred.

6. **Category assignment and EU alignment:** create `sod:CategoryAssignment` resources linking a version to a category concept, and add `skos:exactMatch` to EU Data Theme for aligned categories [7].

7. **Provenance:** record template parsing and mapping as `prov:Activity`, including timestamps and references to source artifacts [8].

8. **Serialization:** publish both RDF Turtle and JSON-LD serializations for integration and tooling compatibility.

## 7. Validation with SHACL

To make the example reproducible beyond a narrative, we specify SHACL shapes that validate the core constraints:

- **OrganVersion constraints:** must have exactly one linked organization, a validity interval, a valid enum value for `organType`, and a spatial value.
- **Temporal constraints:** beginning must not be after end for each validity interval.
- **ChangeEvent constraints:** must have at least one original and one resulting organization, and a timestamp. If `directionInferred` is true, an `inferenceNote` is required.
- **Category alignment constraints:** if an assignment is marked as aligned, the category concept must carry an `skos:exactMatch` to an EU Data Theme concept.

These constraints support an auditable pipeline: violations become explicit, and the validation report can be published alongside snapshots and mappings.

## 8. Planned evaluation and competency questions

We propose evaluating the resulting graph along four dimensions:

- **Coverage:** fraction of organizations and versions that can be represented with complete required fields and validated by SHACL.
- **Temporal fidelity:** ability to represent changes as explicit events and to query organizational history by interval boundaries.
- **Alignment quality:** clarity and traceability of mappings to EU Data Theme, including provenance of mapping rules.
- **Reproducibility:** ability to re-run the transformation and reproduce identical IRIs and statements for a fixed input snapshot.

Example competency questions include:

- Which organizational unit was responsible for datasets in category *Energie* during 2017 to 2021 in a given federal state?
- Which federal states published datasets regarding environment within a specified time interval?
- For a mapped EU Data Theme (for example `eu:ENER`), which federal states show increasing or decreasing publication activity over time once portal categories are aligned?
- Which predecessor/successor relations are inconsistent with validity intervals, and where was a direction inferred from temporal information?

## 9. Conclusion

This position and vision paper argues for treating portal categories and organizational structures as time-scoped knowledge graph resources to enable reproducible longitudinal analyses across Germany's federal Open Government Data landscape. In addition to the conceptual architecture, we provide a worked example derived from a tabular template that demonstrates how organization versions, change events, and EU-aligned category assignments can be represented in RDF and validated with SHACL. The next step is to apply the same design principles to portal-scale extraction snapshots, publish the mapping rules and provenance records, and evaluate the approach on a multi-state sample using competency-question driven analysis.

## A. GitHub Repository for the first workflow

To support reproducibility, we publish the worked example knowledge graph and its accompanying JSON-LD, SHACL shapes, and lightweight ontology in an open GitHub repository: https://github.com/SODIC-research/SODORG. The example corresponds to the RDF pattern in Section 5.2.

## Declaration on Generative AI

During the preparation of this work, the authors used DeepL for text translation and language refinement and ChatGPT 5.2 for code suggestions. After using these tools, the authors reviewed and edited the content as needed and take full responsibility for the publication's content.

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
