# OpenReview forum: "From Federal Heterogeneity to Reproducible Analytics: A Provenance-Aware Knowledge Graph for Cross-Portal Comparability of German Open Government Data"
_eswc-conferences.org/ESWC/2026/Workshop/KGCW — KGCW 2026_

### Official Review · ~Davan_Chiem_Dao1 · 2026-03-19
**Review of "From Federal Heterogeneity to Reproducible Analytics: A Provenance-Aware Knowledge Graph for Cross-Portal Comparability of German Open Government Data"**

**Rating:** 7
**Confidence:** 4

**Review:**

## Summary

The paper addresses the challenge of comparing Open Government Data portals across German federal states, where organizational structures and portal categories differ and evolve over time. It proposes a temporal and provenance-aware knowledge graph that models stable organizations, time-scoped versions, change events, category assignments, and mapping provenance using existing semantic web standards such as ORG, PROV-O, OWL-Time, SKOS, RML, and SHACL. The contribution is mainly presented as a reusable construction pattern illustrated through an example.

## Strengths

- The paper is well written and resources/examples are made available.
- The use case is relevant and well motivated. It is a typical knowledge graph construction use case
- The overall architecture is coherent and technically sound, relying on state-of-the-art standards and components.

## Limitations and Recommendations

- The main limitation is novelty. In my opinion, the proposed construction pattern does not clearly differ from the state of the art, since it mainly combines well-established standards rather than introducing a new method.
- The notion of "stable organization" is unclear/under-specified. It is not fully clear whether this refers simply to an organization that has multiple temporal versions, or to a more abstract notion based on institutional continuity or purpose. In particular, the paper should clarify how this notion behaves in cases such as mergers or splits.
- The minting of version IRIs seems problematic when end dates are encoded in the identifier. If the current version is identified only with its start date at first, it may later need to be renamed to include its end date once that becomes known. This weakens identifier stability and should be clarified.

## Overall Assessment

Overall, this is a clear paper with a very good use case and a coherent technical design. However, its novelty appears limited, since the contribution mainly lies in assembling existing state-of-the-art components for this specific setting. That said, it still provides a good example of a use case that combines many standard semantic web technologies in a meaningful way.

### Typos

Section 6: "hierachicalOrganSuperior" / "hierachicalOrganSuberior" -> hierarchy is missing a r.

---

### Official Review · ~Giorgos_Flouris2 · 2026-03-27
**Interesting ideas, but immature and unclear implementation and presentation**

**Rating:** 5
**Confidence:** 3

**Review:**

This is a paper aiming to produce a Knowledge Graph (KG) regarding German Open Government Data. To the best of my understanding, the main novelty of the paper is that it explicitly aims to tame the inherent dynamicity of the main actors of the domain, e.g., when ministries (or other data producers) merge, split, change names or are reorganized in any manner. The latter creates the need to record the temporal validity of information, along with information regarding the "transitions" (changes) undergone by the entity that each URI refers to.

The ideas of the authors are very good, and the approach is interesting. The aim to represent evolving entities is a valid one, and could prove useful in many domains (most KG creators tend to ignore this aspect, but it is an issue even in less dynamic domains).

Having said that, the paper seemed rather immature to me. Many ideas and concepts are either unclearly presented or not fully developed yet. No evaluation exists, and the exact scope and contents of the resulting KG remain unclear to me. These deficiencies are partly justified by the paper's classification as a "position and vision paper", and it is clear that this is work-in-progress. However, some ideas seem to be not clearly thought-out yet, so it remains questionable whether this will actually work in practice. A bit more maturity is required, in my opinion, to make this paper acceptable. More specific comments appear below.

What is "GND"? I had to look it up, and it probably refers to a standard by the German national library, but the authors should explain that.

The interplay between the so-called "stable" identifiers and the change events is not clear. For example, conside an entity called "X and Y" (e.g., "ministry of education and research") that the state decides to split into two distinct entities "X" and "Y". The original stable identifier would refer to "X and Y", but after the change, would it refer to "X" or to "Y"? If not, what would be the "stable" identifiers of the new entities? And what will happen to these identifiers if the entities are subsequently merged again? More complex examples can be devised, of course. Some clarifications are needed here.

As a more general comment, it is unclear what types of "change events" are allowed. Is there a list? Are there any semantics to it?

Further, the actual contents of the KG are unclear. The paper focuses on the representation of organizations in the German federal states, but once these are modelled, what else will there be in the KG?

The presentation method of the paper is quite "sparse" with lots of headings and subheadings, bullets, and some less-than-important text (e.g., Appendix B could be shortened, part of Listing 1 could be dropped, etc). The authors could easily amend this to fit some additional information that could help the reader better understand the idea.

In conclusion, I believe this is a valid and very useful idea that could find applications in other domains too, and I encourage the authors to continue this effort. However, the work seems rather immature for publication in its current stage.

---

### Official Review · ~Beatriz_Esteves1 · 2026-04-07
**Nice position paper; very early-stage results**

**Rating:** 6
**Confidence:** 4

**Review:**

This position paper proposes a temporal and provenance-aware KG-based framework to integrate heterogeneous public-sector data portals (e.g., different federal states). It emphasizes the modeling of portal-specific schemes as versioned entities, including information on data provenance and its temporal evolution for auditability and accountability of the involved entities. Since the work is framed as a vision/position paper, its contributions are focused on the methodology to build such a framework and not on achieved results. In terms of pros, the article:
- Addresses a real-world issue, i.e., heterogeneous data portals
- Has a strong emphasis on provenance and reproducibility
- Introduces temporal and provenance-aware KG modeling
- Includes evaluation plan
- Provides an openly available repository with RDF examples and corresponding SHACL shapes

However, some limitations can also be described:
- Although the authors address an important issue, it is unclear how it extends to other countries beyond Germany. Furthermore, the authors could also explore if there are lessons learned from other EU Member States' deployments that should be considered for their own methodology
- Limited technical depth, e.g., lacks formalisation, related to the foreseen KG construction framework to be used
- No clear comparison with existing KG integration frameworks (also connected with the first-mentioned limitation)
- Some acronyms are not introduced before the authors use them, e.g., GND

Overall, I believe this paper, as a vision on how to tackle the issue of heterogeneous data portals at the federal level, would contribute to nice discussions at the workshop. However, it must be acknowledged that it is still quite at an early-stage.

---

### Decision · Program_Chairs · 2026-04-09

**Decision:**

Accept

**Comment:**

This paper has been selected for presentation at the KGC workshop. We strongly encourage the authors to consider the reviews whilst revising the paper. Camera-ready instructions will soon follow.